# Anaesthesia care providers employed in humanitarian settings by Médecins Sans Frontières: a retrospective observational study of 173 084 surgical cases over 10 years

Søren Kudsk-Iversen [ID],[1] Miguel Trelles,[2] Elie Ngowa Bakebaanitsa,[2,3] Longin Hagabimana,[2,4] Abdul Momen,[2,5] Rahmatullah Helmand,[2,6] Carline Saint Victor,[2,7] Khalid Shah,[2,8] Adolphe Masu,[2,4,5,9] Judith Kendell,[2] Hilary Edgcombe,[1] Mike English[10,11]

For numbered affiliations see end of article.

**Correspondence to**
Dr Søren Kudsk-Iversen;
sorenki@gmail.com

## ABSTRACT

**Objective** To describe the extent to which different categories of anaesthesia provider are used in humanitarian surgical projects and to explore the volume and nature of their surgical workload.

**Design** Descriptive analysis using 10 years (2008–2017) of routine case-level data linked with routine programme-level data from surgical projects run exclusively by Médecins Sans Frontières-Operational Centre Brussels (MSF-OCB).

**Setting** Projects were in contexts of natural disaster (ND, entire expatriate team deployed by MSF-OCB), active conflict (AC) and stable healthcare gaps (HG). In AC and HG settings, MSF-OCB support pre-existing local facilities. Hospital facilities ranged from basic health centres with surgical capabilities to tertiary referral centres.

**Participants** The full dataset included 178 814 surgical cases. These were categorised by most senior anaesthetic provider for the project, according to qualification: specialist physician anaesthesiologists, qualified nurse anaesthetists and uncertified anaesthesia providers.

**Primary outcome measure** Volume and nature of surgical workload of different anaesthesia providers.

**Results** Full routine data were available for 173 084 cases (96.8%): 2518 in ND, 42 225 in AC, 126 936 in HG. Anaesthesia was predominantly led by physician anaesthesiologists (100% in ND, 66% in AC and HG), then nurse anaesthetists (19% in AC and HG) or uncertified anaesthesia providers (15% in AC and HG). Across all settings and provider groups, patients were mostly healthy young adults (median age range 24–27 years), with predominantly females in HG contexts, and males in AC contexts. Overall intra-operative mortality was 0.2%.

**Conclusion** Our findings contribute to existing knowledge of the nature of anaesthetic provision in humanitarian settings, while demonstrating the value of high-quality, routine data collection at scale in this sector. Further evaluation of perioperative outcomes associated with different models of humanitarian anaesthetic provision is required.

### Strengths and limitations of this study

► This is the largest study detailing how anaesthetic task sharing and shifting is employed in the humanitarian sector.

► Additionally, we believe this is the first study to describe the extent of the presence and caseload of uncertified anaesthetic providers in humanitarian surgical projects.

► Due to the nature of the linked data, we were unable to connect anaesthetic provider with individual operations.

► Therefore, to limit the misclassification bias, we do not ascribe a provider to each case, but rather describe the most senior provider available in the surgical project (the 'anaesthetic lead').

## INTRODUCTION

Globally, there is a large unmet surgical need. Low-income and middle-income countries (LMIC) are disproportionately affected by gaps in healthcare provision, with an estimated 90% of patients in these countries unable to access basic surgical care.[1] The burden is increased and access further reduced in crisis situations, caused by conflict or natural disasters.[2] To address these imbalances, Médecins Sans Frontières (MSF, also known as Doctors without Borders) provide humanitarian surgical assistance based on the needs of affected populations through one or more of their five operational centres, one of which is Operational Centre Brussels (MSF-OCB).

There is an increasing body of literature outlining the surgical needs of populations in humanitarian settings.[3–6] The recognition that the humanitarian sector is not immune

from the need to demonstrate safe surgical care has led to calls for more robust outcome data and clearer accountability.[7–9] Only few studies, limited by small study size and limited external validity, have addressed the composition of the surgical workforce employed by humanitarian organisations.[10 11] Therefore, there is inadequate published data on whether different anaesthesia providers (eg, physician, nurse or other healthcare provider) are employed in different settings, and to what extent there is a physician expatriate presence within the team. In order to comment on outcomes and identify areas where practice can be improved, it is essential to know who provides the care and if there is any learning that can be derived from their practice.

The objective of this study is to describe the extent to which different categories of anaesthesia provider are used in humanitarian surgical projects and to explore the volume and nature of their surgical workload.

## METHODS

The study fulfilled the exemption criteria set by the MSF Ethics Review Board (ERB) for a posteriori analyses of routinely collected clinical data and thus did not require MSF ERB review. It was conducted with permission from Medical Director, MSF-OCB. This exemption did not allow country-specific/site-specific detail to be included,

therefore we aggregate data within the WHO regional groupings.[12]

The findings are reported in accordance with the REporting of studies Conducted using Observational Routinely collected health Data (RECORD) statement, the extended Strengthening the Reporting of Observational Studies in Epidemiology (STROBE) statement on routinely collected data.[13]

### Study design

This was a descriptive study of routine data collected between January 2008 and December 2017. We excluded any incomplete data and data from surgical projects where MSF-OCB were collaborating with other MSF operational centres or local governments, as we were unable to account for workforce or resources made available by others than MSF-OCB.

We linked three sources of data (figure 1): 1) case-level routine surgical surveillance data were recorded by theatre staff in logbooks on-site, then transcribed onto an Excel spreadsheet and finally transferred to Brussels on a monthly basis where they were reviewed and any missing or extraneous data were queried with the local teams; 2) programme-level data, available from MT (head of the Surgical, Anaesthesia, Gynaecology and Emergency Medicine unit during this period) were reviewed; 3) end of deployment reports written by expatriate physician

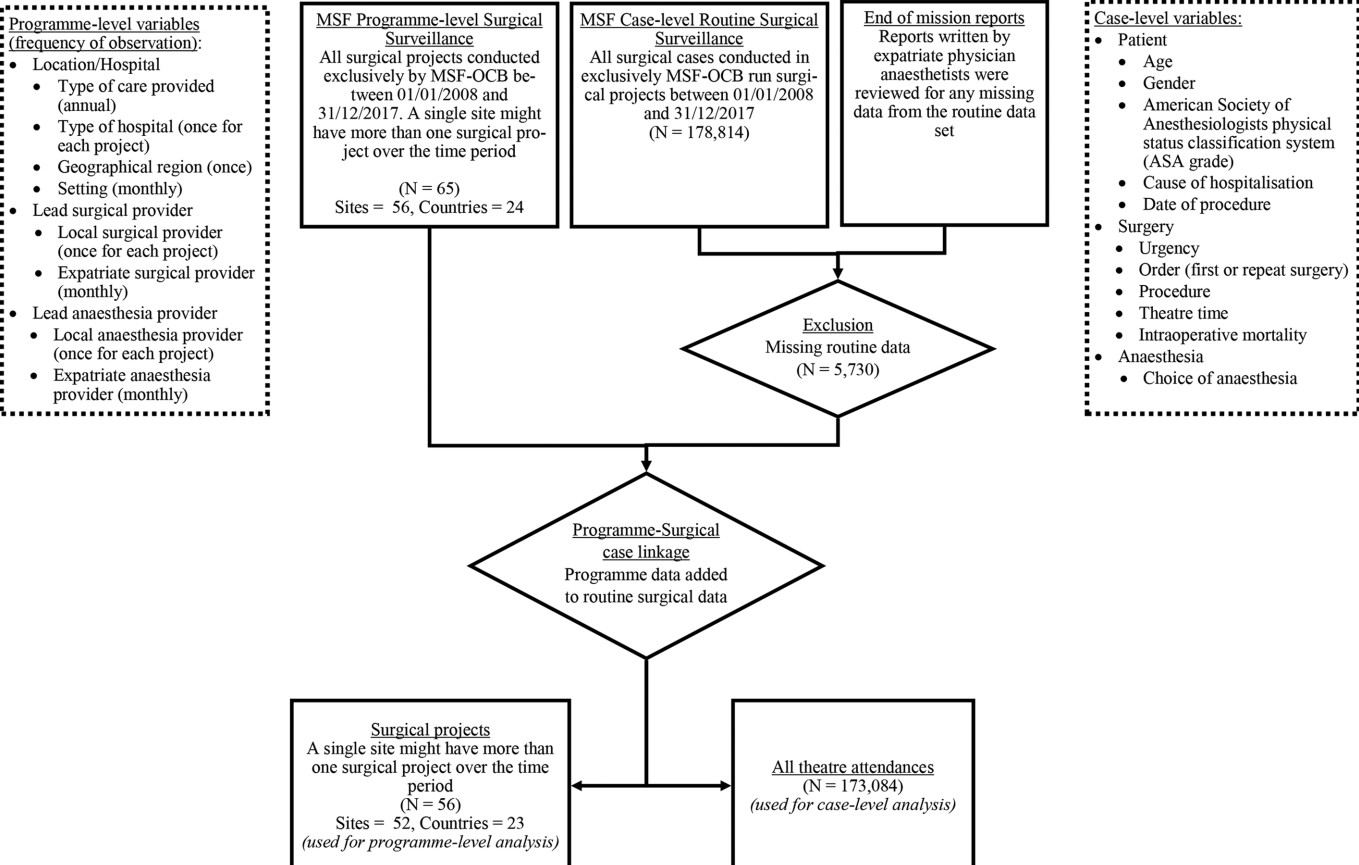

**Figure 1** Flow diagram showing inclusion/exclusion of data and points of data linkage. MSF-OCB, Médecins Sans Frontières-Operational Centre Brussels.

anaesthesiologists were reviewed to fill gaps in data from the case-level data. Data were deidentified at point of data collection, and were only accessed by SK-I, MT and JK. Any data shared with the remaining coauthors were fully anonymised.

## Setting and anaesthesia providers

Three different project setting types were identified: 1) regions recently affected by sudden-onset natural disasters (ND), where MSF deployed an entire expatriate surgical team in accordance with WHO minimum requirements,[14] 2) active armed conflict (AC) situations and 3) stable situations where MSF supported a pre-existing local facility to address healthcare gaps (HG), which existed for a variety of reasons, including the aftermath of natural disasters or armed conflict.

The setup and duration of surgical projects varied. Some projects were intended to operate only for a short period, either within existing local infrastructure or through fully self-contained surgical platforms. Other projects were set up to serve for a longer period or evolved over time into a fully functioning hospital with ability to provide complex care provision. The different hospital types are described in detail in the online supplementary appendix table 1. The setup was not dictated by the setting, and could change over the course of a project.

During the 10-year study, anaesthesia provision was led by one of the following: a) specialist physician anaesthesiologists, either local or expatriate (from both high-income and low-income settings) doctors with qualifications in anaesthesia, b) nurse anaesthetists, either local or expatriate (predominantly from low-income settings) nurses or other non-physician clinical cadres with formal training and qualification in anaesthesia in their country of origin or c) uncertified anaesthesia providers, local nurses or allied healthcare professionals with a broad range of different levels of experience in anaesthesia provision but without a formal qualification who received on-the-job training only. The MSF-OCB anaesthesia referent assesses the provider requirement for each location based on expected workload, job description and staff availability. For example, if a project is expected to have a low workload, nurse anaesthetists are either recruited locally or, if they are senior providers, sent over as expatriates from MSF-OCB surgical projects in other countries. In situations where MSF-OCB are unable to source qualified staff for a surgical project, they may hire the existing local uncertified anaesthesia providers, who will all receive on-the-job training by MSF and supervision by expatriate physician anaesthesiologists for a trial period. These situations should result in uncertified anaesthesia providers working in settings with a low workload and with distant supervision available from a nearby hospital with MSF-OCB involvement where anaesthesia is led by an expatriate physician anaesthesiologist. All MSF surgical projects have standardised anaesthetic equipment and medications, as described elsewhere.[5]

## Variables and bias

Different variables were retrieved from the three different data sources. From the routine case-level data (and end of deployment reports), we identified patient variables (including age and sex), surgical and anaesthetic variables (including type of surgery, type of anaesthesia) and geographic location of the cases done. From the programme-level data, we obtained additional surgical and anaesthetic variables (including provider level of training, presence of expatriate), and location variables (including project setting, type of hospital). A detailed description of all variables used is available in the online supplementary appendix table 1.

The use of routine surveillance data puts the study at risk of selection bias, which may risk under-reporting by some providers (eg, expatriates visiting for short periods who may be unfamiliar with the data collection tool, or staff who for whatever reason choose not to document cases) or in busy settings (eg, high workload or strained workforce). While we cannot account for surgical cases not recorded in the first place, we explored incomplete data that had been excluded to assess similarity to the included data.

Furthermore, it should be noted that provider data were available showing the most senior provider present for each project, not per case (and for expatriates, was updated monthly during a project). This puts the study at risk of misclassification bias regarding the anaesthesia providers in favour of the most senior team member regardless of their presence in theatre. Additionally, it would be easy to over-represent the case-level involvement of physician anaesthesiologists (especially when they are present as expatriates, as they might be more restricted in their movement and have additional non-clinical commitments). We therefore present data according to the most senior provider present on the project in a given month (the anaesthetic 'lead'). We also note which projects had a visiting expatriate physician anaesthesiologist present.

## Statistical analysis

Data were collected and linked in Excel (2016) and data cleaning and analysis was performed in R V.3.6. Continuous data were assessed for normality, and no parametric data were identified. For non-parametric continuous and numeric ordinal data, median (IQR) and full range were reported. For categorical variables, the raw counts were reported.

We stratified our analysis according to the settings identified, as they might influence the extent and pattern by which different anaesthetic providers were deployed.

However, data from surgical projects in the WHO South East Asia region and in ND settings were described separately due to their small numbers and being separate from the dominant regions (online supplementary appendix tables 2 and 3).

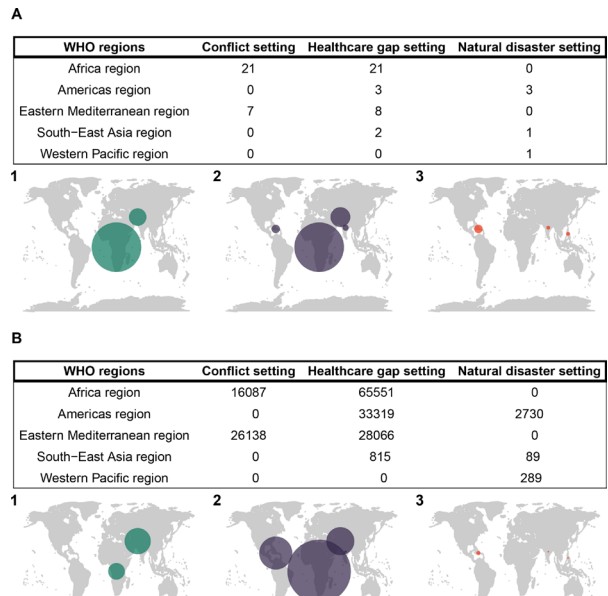

**Figure 2** World maps showing number of (A) surgical projects and (B) surgical cases in each WHO region in settings of (1) conflict, (2) healthcare gaps and (3) natural disasters.

## Patient and public involvement

There was no involvement of patients or the public in the development or execution of this study.

## RESULTS
### General findings

Over the 10 years, a total of 173 084 cases had full routine data collected (96.8% of all cases) across 23 countries and 52 different locations (figure 1). The majority of cases occurred in HG settings, and in the WHO Africa region

(figure 2). Surgical projects in settings of ND represented 3108 cases (<2% of the total number of operations over the time period) and a total duration of 40 project-months over five sites; anaesthesia care in the ND setting was exclusively led by physician anaesthesiologists (online supplementary appendix table 3).

Overall, the shortest surgical project lasted a month, and the longest lasted beyond the 10 years covered by this study (figure 3). Surgical projects in HG settings stayed open for longer (median 866 days, IQR 360.25–1900 days) than projects in AC and ND settings (287.5, 173–498.25 days and 210, 122–308 days, respectively). The workload within each project varied widely, with 31 projects accounting for 5.1% of all cases, and four projects accounting for 47.6% (figure 3A).

Of the four biggest projects, anaesthesia for two projects was exclusively physician anaesthesiologist-led (one in the WHO Eastern Mediterranean region in an AC setting, the other in the WHO Americas region in a HG setting). The third project was predominantly physician anaesthesiologist-led (in the WHO Eastern Mediterranean region) progressing from an initial AC to become a stable HG setting. The last was predominantly uncertified anaesthesia provider-led with a periodic presence of expatriate physician anaesthesiologists (in the WHO Africa region, starting in AC and then becoming a stable HG setting). Data for these four major projects followed a similar pattern of distribution (in terms of case and programme-level data) to the remaining dataset of all other projects, and have therefore been included in the findings below.

## Programme-level provider findings

Most surgical projects (23/28 in AC, 25/32 in HG and all 5 in ND) included a period of anaesthesia provision led by physician anaesthesiologists (figure 3B and

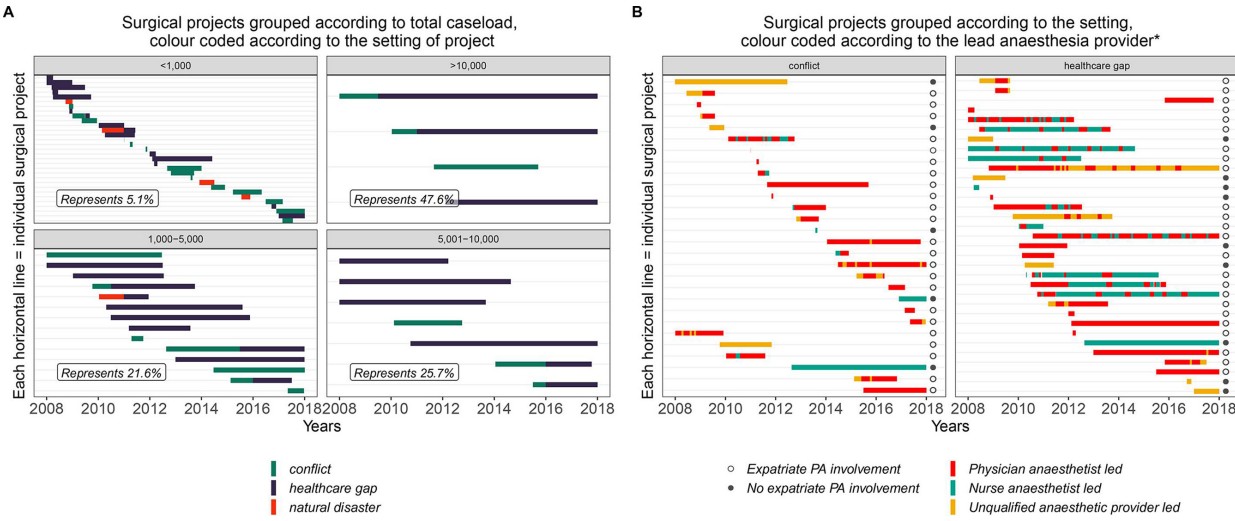

**Figure 3** Timelines showing when the included surgical projects were active and their duration.
*Excludes periods where projects are run in collaboration with other organisations or local government. Additionally, only data from 2008 till 2017 are included. Therefore, periods with expatriate physician anaesthetist (PA) involvement before then are not reflected here.

table 1). Anaesthesia in any setting with sole trauma care was mostly led by physician anaesthesiologists (table 1). If anaesthesia provision in a project was not fully physician anaesthesiologist-led, the pattern of their presence in most cases involved short periods (usually around 3 months) over the course of the surgical project, mostly towards the start of the project (figure 3B). Overall, a physician anaesthesiologist was identified as present for 737 (49%) project-months in AC and HG (table 1). However, in these settings >66% of cases overall were conducted during periods where physician anaesthesiologist were present in the projects (80% of cases in AC and 60% of cases in HG).

When there was not a physician anaesthesiologist attached to a project, anaesthesia was most commonly led by nurse anaesthetists in the HG setting and most commonly led by uncertified anaesthesia providers in the AC setting.

### Case-level provider findings

Case-mix was similar across all lead providers with respect to age (mostly young adults) and underlying health (mostly ASA 1) (table 2). All providers did predominantly non-elective work with trauma surgery more commonly done in physician-led projects in both AC and HG and caesarean sections more commonly done in nurse anaesthesia projects, especially in HG settings. The intraoperative mortality was 0.3% and 0.3% in physician anaesthesiologist-led project-months, 0.2% and 0.1% in nurse anaesthetist-led project-months and 0.3% and 0.2% in uncertified anaesthesia provider-led project-months in AC and HG settings, respectively.

All lead providers made use of the two most common types of anaesthesia: spinal injection alone and general anaesthesia (GA) without intubation or muscle relaxant, which for the most part was ketamine-based. This was done in broadly similar proportions when comparing surgical categories in different settings (eg, spinal injection and GA without protected airway for caesarean section was 61%–70% and 22%–36%, respectively in AC, and 78%–86% and 6%–14%, respectively in HG).

### Missing data

The cases excluded due to missing variables (5730, 3.2%) are predominantly from the early years. The three most common variables with missing data were ASA score (3232 missing), intraoperative mortality (2154) and time in theatre (1922) (see appendix, missing data, online supplementary appendix table 1). The data with missing intraoperative mortality was exclusively from 2008, and were predominantly from two projects in the WHO Africa region where the bulk of the work was elective surgery for training purposes. Eight surgical projects were completely excluded (seven in healthcare gap settings, one that was in both natural disaster settings and healthcare gap settings, see appendix, missing data, online supplementary appendix figure 1), all with a caseload of <100 operations and a short period of activity. The missing

data were predominantly from projects with uncertified anaesthesia provider-led or physician anaesthesiologist expatriate-led provision. This suggest the data were not missing completely at random and may risk introducing bias, although they comprised a small overall proportion of cases and available variables suggest the excluded cases were similar to the analysed dataset (see appendix, missing data, online supplementary appendix table 2).

## DISCUSSION

This is the largest observational study published from a humanitarian organisation describing the types of anaesthesia providers employed and the pattern of their work in a number of different settings. While not all humanitarian organisations (and MSF operational centres) operate in the same way as MSF-OCB, this study provides useful insights that may contribute towards their operational strategies.

Over 10 years of surgical activity by MSF-OCB, we found that anaesthesia provision was led by physician anaesthesiologists during 66% of all cases in HG and AC settings (bearing in mind physician anaesthesiologist-led does not mean physician anaesthesiologists administered the anaesthesia) with nurse anaesthetist-led provision accounting for 19% and uncertified anaesthesia provider-led provision accounting for 15% of cases. There was some variation in the surgical caseload between provider types: physician anaesthesiologists were more commonly attached to projects with trauma-related surgery, while nurse anaesthetists were more commonly the most senior anaesthetic provider in projects with high numbers of obstetric surgery. All providers led during surgery on both very sick (ASA grade 5) and very young patients (aged only a few days), although majority of cases were minor surgery, which are less risky even in patients with a higher ASA class. In locations with uncertified anaesthesia provider-led provision, which was predominantly in the WHO Africa region, there was also a reduced presence of specialised surgical providers and expatriate involvement, despite the patient profile and surgical caseload being largely similar to that encountered in physician anaesthesiologist-led surgical projects in similar settings.

MSF tries to avoid employing uncertified anaesthesia providers, and they continue to evaluate means of mitigating this risk. However, a set of unique circumstances makes it unavoidable on occasion: 1) MSF, like many humanitarian organisations, operate predominantly in locations where there is a pre-existing anaesthesia workforce shortage,[15] and often in situations where this shortage may be exacerbated due to armed conflict or population displacement; 2) expatriate staff are not always available, as MSF only deploy senior qualified anaesthesiologists as their expatriates, and it may not be possible for them to take time away from work at short notice; 3) even if expatriate staff are available, in many contexts they have become deliberate targets. This has

**Table 1** Programme-level descriptive table according to number of surgical projects and month of activity in different settings

### A: Programme-level descriptive table according to number of surgical projects in different settings

| | Conflict (surgical projects=28) | | | Healthcare gap (surgical projects=32)* | | |
|---|---|---|---|---|---|---|
| | Physician anaesthesiologist-led | Nurse anaesthetist-led | Uncertified anaesthetic provider-led | Physician anaesthesiologist-led | Nurse anaesthetist-led | Uncertified anaesthetic provider-led |
| Number of surgical projects involved in at any point† | 23 | 14 | 12 | 25 | 17 | 12 |
| Type of hospital‡ in surgical project, number of surgical projects involved in at any point§ | | | | | | |
| Sole remit hospital | 5 | 2 | 1 | 13 | 7 | 1 |
| Referral hospital | 6 | 5 | 2 | 7 | 5 | 2 |
| District hospital | 11 | 4 | 8 | 7 | 4 | 7 |
| Health centre | 1 | 3 | 1 | 0 | 1 | 2 |
| Type of surgical care performed in project, number of surgical projects involved in at any point§ | | | | | | |
| Emergency only | 9 | 8 | 5 | 3 | 4 | 4 |
| Capacity to perform both emergency and elective surgery | 8 | 4 | 6 | 11 | 5 | 6 |
| Maternity care only | 1 | 1 | 1 | 9 | 6 | 1 |
| Trauma care only | 4 | 1 | 0 | 2 | 0 | 0 |
| Other specific care provision¶ | 1** | 0 | 0 | 2†‡‡ | 1§§ | 1¶¶ |

### B: Programme-level descriptive table according to months of activity in different settings

| | Conflict (surgical projects=28) | | | | Healthcare gap (surgical projects=32)* | | | |
|---|---|---|---|---|---|---|---|---|
| | Physician anaesthesiologist-led | Nurse anaesthetist-led | Uncertified anaesthetic provider-led | All conflict missions | Physician anaesthesiologist-led | Nurse anaesthetist-led | Uncertified anaesthetic provider-led | All healthcare gap missions |
| Number of months active in any mission | 235 | 75 | 94 | 404 | 502 | 429 | 160 | 1091 |
| Surgical provider, number of months present (% of cohort) | | | | | | | | |
| General and specialty surgeon | 100 (43) | 13 (17) | 0 (0) | 113 (28) | 260 (52) | 105 (24) | 3 (2) | 368 (34) |
| General surgeon only | 96 (41) | 58 (77) | 20 (20) | 174 (43) | 115 (23) | 133 (31) | 38 (24) | 286 (26) |
| Specialty surgeon only | 31 (13) | 2 (3) | 4 (4) | 37 (9) | 103 (21) | 162 (38) | 7 (4) | 272 (25) |
| MD | 8 (3) | 2 (3) | 70 (74) | 80 (20) | 24 (5) | 29 (7) | 112 (70) | 165 (15) |

*South East Asia region contributed such a small proportion to missions in 'healthcare gap' settings (2 missions, 815 cases or <1%), that they have been excluded and instead described in online supplementary appendix table 2.
†Surgical projects can have anaesthesia provision by multiple different providers during the period they are open. Therefore, the rows might add up to more than the total number of projects in each setting.
‡Definitions of hospitals found in online supplementary appendix table 1.
§Two surgical projects changed from being able to provide both emergency and elective surgery, to providing solely maternity care. As such, they are counted twice under 'type of hospital' and 'type of surgical care'.
¶Specific care provision are surgical projects with a specific care remit.
**Wound care.
††Trauma and surgical care.
‡‡Obstetric fistula care.
§§Surgical care of typhoid-related complications.

**Table 2** Case-level descriptive table grouped according to setting*

| | Conflict (surgical projects=28, n=42 225) | | | Healthcare gap (surgical projects=32, N=126 936)† | | |
|---|---|---|---|---|---|---|
| | Physician anaesthesiologist-led | Nurse anaesthetist-led | Uncertified anaesthetic provider-led | Physician anaesthesiologist-led | Nurse anaesthetist-led | Uncertified anaesthetic provider-led |
| Number of all surgical episodes | 33 763 | 3798 | 4664 | 78 126 | 28 559 | 20 251 |
| Patient demographics | | | | | | |
| Female, no. (%) | 12 424 (37) | 1888 (50) | 2237 (48) | 38 919 (50) | 22 439 (79) | 12 834 (63) |
| Median age, years (IQR, (range)) | 23 (15–33, (1 day–105)) | 25 (16–34, (2 days–90)) | 23 (18–30, (3 days–94)) | 28 (19–37, (1 day–102)) | 26 (20–34, (1 day–98)) | 25 (16–35, (1 day–96)) |
| ASA, value (IQR, (range)) | 1 (1–2, (1–5)) | 2 (1–2, (1–5)) | 1 (1–2, (1–5)) | 1 (1–2, (1–5)) | 1 (1–2, (1–5)) | 1 (1–2, (1–5)) |
| Cause of hospitalisation, no. %: | | | | | | |
| Trauma (intentional or unintentional) | 21 968 (65) | 1384 (36) | 2366 (51) | 42 454 (54) | 2850 (10) | 6303 (31) |
| Obstetric | 6642 (20) | 1073 (28) | 1295 (28) | 20 387 (26) | 18 270 (64) | 7211 (36) |
| Other‡ | 5153 (15) | 1341 (35) | 1003 (22) | 15 285 (20) | 7439 (26) | 6737 (33) |
| Surgical demographics | | | | | | |
| Urgency, no. (%) | | | | | | |
| Emergent | 14 344 (42) | 2203 (58) | 2581 (55) | 37 234 (48) | 19 922 (70) | 9221 (46) |
| Urgent | 18 091 (54) | 1115 (29) | 1961 (42) | 34 771 (45) | 4781 (17) | 8699 (43) |
| Elective | 1328 (4) | 480 (13) | 122 (3) | 6121 (8) | 3856 (14) | 2331 (12) |
| Proportion of cases from initial presentation, n (%) | 20 079 (59) | 3053 (80) | 3588 (77) | 51 493 (66) | 25 597 (90) | 14 492 (72) |
| Median time in theatre, minutes (IQR, (range)) | 50 (30–70, (7–710)) | 50 (35–70, (15–356)) | 45 (35–65, (10–360)) | 60 (35–90, (10–870)) | 60 (50–80, (10–1140)) | 50 (30–70, (5–460)) |
| Main categories of surgery, no. (%)§ | | | | | | |
| Minor surgery | 20 670 (61) | 1688 (44) | 3019 (65) | 37 419 (48) | 6798 (24) | 9906 (49) |
| Caesarean section | 4758 (14) | 891 (23) | 884 (19) | 16 138 (21) | 13 336 (47) | 6259 (31) |
| Visceral surgery | 3709 (11) | 949 (25) | 555 (12) | 11 109 (14) | 4856 (17) | 2922 (14) |
| Orthopaedic surgery | 3372 (10) | 90 (2) | 48 (1) | 9408 (12) | 151 (1) | 328 (2) |
| Obstetric and gynaecological surgery (excluding caesarean section) | 802 (2) | 122 (3) | 139 (3) | 3226 (4) | 3147 (11) | 756 (4) |
| Specialties¶ | 452 (1) | 58 (2) | 19 (0) | 826 (1) | 271 (1) | 80 (0) |
| Intraoperative mortality, no. (%) | | | | | | |
| For all cases | 102 (0.3) | 7 (0.2) | 16 (0.3) | 204 (0.3) | 31 (0.1) | 31 (0.2) |

*Percentages have been rounded to nearest full digit, and might not add up to 100%.
†South East Asia region contributed such a small proportion to missions in 'healthcare gap' settings (2 missions, 815 cases or <1%), that they have been excluded and instead described in the online supplementary appendix table 2.
‡ 'Other' causes of hospitalisation include: tropical disease related, tumours, non-tumour-related obstruction and complications from traditional medical practices.
§The surgical procedures included in each grouping can be found in the online supplementary appendix table 4.
¶Specialties encompass (total number of cases across whole dataset): urology (726), vascular surgery (355), plastic and reconstructive surgery (144), ENT surgery (116), neurosurgery (115), surgery within thoracic cavity (108), maxillofacial surgery (61) and other forms of specialised surgical care that does not fall into the aforementioned categories (109).

led to more cautious deployment of expatriate personnel into volatile settings.[16]

In this study, we report briefly on intraoperative mortality. Rates are comparable across the different lead providers and similar to other observational data from LMICs[17–21] and some humanitarian organisations (including other MSF operational centres),[4 22 23] while higher than other humanitarian organisations.[24 25] Such data must be interpreted cautiously as they should ideally be adjusted more fully for case-mix and severity. Furthermore, most mortality related to surgery occurs in the days following surgery and not in theatre,[23 26 27] and these data are not available as part of the routine data we analysed. While a more appropriate and widely recognised measure of surgical outcomes is perioperative mortality, which is advocated by both the Lancet Commission on Global Surgery and WHO,[28 29] we were unable to report this. Further research into surgical outcomes in the humanitarian setting, which includes perioperative mortality and the incidence of postoperative complications and how they might differ between different anaesthesia providers, would be useful to assist organisations in providing safe and efficient anaesthesia in resource-limited situations.

## Limitations

Data quality is a known issue when using surveillance data, and the occasionally unpredictable nature of working in humanitarian settings means there is a risk of further decline in quality. Due to the rigour in data monitoring centrally by MSF-OCB on a regular basis as described in the 'Methods' section, much has been done to minimise both missing data and improve the quality of the collected dataset. Our approach does have a particular risk of misclassification related to expatriate physician presence. Cases or projects could have been identified as 'physician anaesthesiologist-led', but the physician anaesthesiologist may not actually have been in the operating room for a variety of reasons including overseeing multiple theatres, or curfew and security concerns. Such misclassification could under-represent the proportion of work where non-physicians were effectively sole providers. Our results therefore likely present a conservative estimate of the care provided by nurse anaesthetist and uncertified anaesthesia provider. Finally, it is important to note that some projects had started before the start of routine data collection in 2008. Projects with expatriate physician anaesthesiologists providing on-the-job training for uncertified anaesthesia providers in the period before 2008 will not be reflected in our dataset.

## CONCLUSION

The majority of MSF anaesthesia care is done in teams where there are physician anaesthesiologists available. In conflict and healthcare gap settings, nurse anaesthetists and uncertified anaesthesia providers can be used as major providers. This study shows that the humanitarian sector has considerable experience with task sharing and shifting but further study of perioperative outcomes in these circumstances is needed to draw conclusions about how safe and practical it would be to apply to other settings. Despite their limitations, routine data are key to monitoring the effectiveness of health systems, including humanitarian care, at scale and the MSF-OCB dataset is an important resource demonstrating that valuable data can be collected even in difficult circumstances. There is a need for wider engagement by the humanitarian community to continue to improve the collection and use of valid surgical outcome data. This would promote learning on how to optimise the surgical and anaesthetic workforce and help to ensure safe surgical and anaesthetic care in the humanitarian sector.

**Author affiliations**
[1]Nuffield Department of Anaesthetics, Oxford University Hospitals NHS Foundation Trust, Oxford, UK
[2]Operational Centre Brussels, Médecins Sans Frontières, Bruxelles, Belgium
[3]Masisi Referral Hospital, Masisi—MSF Democratic Republic of the Congo mission, Masisi, The Democratic Republic of the Congo
[4]Arche Trauma Hospital, Bujumbura—MSF Burundi mission, Bujumbura, Burundi
[5]Khost Maternity, Khost—MSF Afghanistan mission, Khost, Afghanistan
[6]Ahmad Shah Baba Hospital, Kabul—MSF Afghanistan mission, Kabul, Afghanistan
[7]Tabarre Trauma Hospital, Port-au-Prince—MSF Haiti mission, Port-au-Prince, Haiti
[8]Timurgara District Headquarter Hospital, Timurgara—MSF Pakistan mission, Timurgara, Pakistan
[9]Castors Maternity, Bangui—MSF Central African Republic mission, Bangui, Central African Republic
[10]Health Services Unit, KEMRI—Wellcome Trust Research Programme, Nairobi, Kenya
[11]Centre for Tropical Medicine and Global Health, University of Oxford, Oxford, UK

**Contributors** SK-I helped conceive the study design, analyse the data, interpret the data, write the initial draft of the manuscript and edit the manuscript. MT helped conceive the study design, collect the data, interpret the data and edit the manuscript. ENB, LH, AM, RH, CSV, KS, and AM helped collect the data and edit the manuscript. JK helped collect the data, interpret the data and edit the manuscript. HE and ME helped conceive the study design, interpret the data and edit the manuscript.

**Funding** SK-I received funding from NIHR through their academic clinical fellowship scheme. ME received funding from a Wellcome Trust Senior Fellowship (#207522) as part of an unrelated research grant.

**Competing interests** All authors except from SK-I, ME and HE are employed by MSF-OCB.

**Patient consent for publication** Not required.

**Ethics approval** The study protocol was submitted to the MSF Ethics Review Board and the Oxford Tropical Research Ethics Committee who granted ethical exemption.

**Provenance and peer review** Not commissioned; externally peer reviewed.

**Data availability statement** All data relevant to the study are included in the article or uploaded as supplementary information. The data used in the study were provided by Dr MT (SAGE Coordinator at MSF-Brussels at the time data were obtained), and contains deidentified case-level routine surgical surveillance data and programme-level data. All relevant data are available in the tables, figures and appendix.

**ORCID iD**

Søren Kudsk-Iversen http://orcid.org/0000-0002-7112-3548

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
