## [Reviewer comments · BMJ Open]

ARTICLE DETAILS

TITLE (PROVISIONAL)	Anaesthesia care providers employed in humanitarian settings by Médecins Sans Frontières: A retrospective observational study of 173,084 surgical cases over 10 years
AUTHORS	Kudsk-Iversen, Soren; Trelles, Miguel; Ngowa Bakebaanitsa, Elie; Hagabimana, Longin; Momen, Abdul; Helmand, Rahmatullah; Saint Victor, Carline; Shah, Khalid; Masu, Adolphe; Kendell, Judith; Edgcombe, Hilary; English, Mike

VERSION 1 – REVIEW

REVIEWER	meena nathan Cherian Geneva Foundation for Medical Education and Research Geneva, Switzerland
REVIEW RETURNED	25-Oct-2019

GENERAL COMMENTS	Page 7, Line 30: PA can be clarified better e.g. Physician Anesthesiologist (specialist). It is not clear in this study if there were Physicians/doctors (non-specialists, MD) providing anesthesia? This is a common scenario in low- and middle-income countries, particularly, 'task sharing' during AC situations, resource limited situations, or unique circumstances as mentioned in page 15, line 42. Page 11, Table 1A: 'MD' has been defined for surgical provider, therefore a similar inclusion (if possible), could be made for anesthesia provider. Line 40: NAs are recruited locally (or sent as expatriates if they are senior providers). Page 12, Line 16: ...and ketamine-based general anaesthesia (GA) without a protected airway. Need to clarify if the above statement is a new variable from what has been identified in Page 30 Appendix, Line 16 Table 1 Variables e.g: General anaesthesia without intubation or muscle relaxant; Other anesthesia. Page 13, Table 2: Clarify 'Emergent' (not mentioned anywhere in the appendix). Page 16, Line 16, 49: Correct/Rephrase these statements:considering the at times unpredictable nature of working.....'learning that' could be valuable in many LMIC... Page 19, Line 12, Fig 1: It is not clear as to what is 'Order' under the title Case levels ?
--

REVIEWER	Lena E Dohlman MD MPH MGH/Harvard Medical School USA
REVIEW RETURNED	20-Nov-2019

GENERAL COMMENTS	Reviewer Comments The use of different anaesthesia providers in humanitarian settings: Descriptive study of 173,084 episodes of surgical care provided by Medecins Sans Frontieres over 10 years This is a descriptive retrospective study of data collected over 10 years that reports the surgical and anaesthetic staffing and types of surgical cases performed in three different humanitarian settings. The three settings were 1. Active natural disasters (ND) 2. Active conflict areas (AC) 3. Areas with health care gaps for other reasons (HG). The data presented was mostly focused on AC and HG settings. Although there was missing data, particularly in mortality and ASA classification of patients, the data that has been collected should certainly be of interest to those involved in humanitarian surgical care and the state of anesthesia provider capacity between 2008 and 2017 in the settings described. The strength of the study is the large amount of data collected under difficult circumstances over a decade. The weaknesses, as the authors pointed out, is that physician anesthesiologist led care does not mean physician anesthesiologist involved care and there was missing and incomplete data in mortality reporting. This makes it difficult to draw conclusions about the safety differences of using various anesthesia providers but doesn't negate the importance of reporting the state of available anesthetic care providers and the relative safety of intraoperative anaesthetic care in MSF settings . Recommendations:  1. Change the title to make it clearer what you are reporting on. One suggestion might be “Anaesthesia care providers employed in humanitarian settings by Medecins Sans Frontieres: A descriptive study of 173,084 surgical cases over 10 years.  2. Consider rewording your objective. Is your primary objective to characterize the volume and nature of the surgical workload or is that important as context only? What do you mean by exploring the “nature” of the use of different categories of anaesthesia providers? Do you mean “to explore the numbers of different levels of training of anesthesia providers used in humanitarian settings” The objective can be made clearer. 3. In the abstract results there is mention of mortality as “low”, please explain compared to what or give the percentage range. 4. The authors mention natural disasters as one of the settings included in the study but only include the data in the appendix. Consider including it for comparison with the AC and HG settings. Despite the smaller numbers, it is helpful to have it together for comparison. 5. Please change the nomenclature. In much of the world “anesthetist or anaesthetist” refers to a non-physician and “anesthesiologist or anaesthesiologist” is a physician. In North America a PA is a physician’s assistant. There is a great deal of confusion among many about the difference between a physician and nurse anesthesia provider- even in the medical community. I recommend that you decrease the confusion by using the full name of physician anaesthesiologist where indicated and non-physician anesthetist or non-certified anaesthetist. By “non-qualified” I believe the authors mean non-certified or unofficially trained. I don't think MSF would have permitted someone unqualified (meaning incompetent) to do the work with MSF.
---

	6. The kind of anaesthesia used for cases is very interesting and should have a more prominent place in the article. The authors currently have it in the “Case-level findings”. 7. Please say more about the 8 surgical projects that were completely excluded if you know more. It would also be interesting to know about the missing intra-operative mortality data. Which settings did they come from and what else do you know about the providers and case mix from those missing the data? I believe the answers may be in the Matrixplot but this is difficult to read. I wonder if you might try dividing it into two figures to make the labels bigger or excluding data that is irrelevant to the reader such as site ID. 8. Instead of writing “physician-led” you might consider writing “physician available team care”. This would make it clearer that you are reporting on care settings where physician anaesthesiologists were available but not necessarily present. 9. I believe your conclusion overstates that you can “inform global debates on the provision of anaesthesia” Perhaps you can “add to the global data on anaesthetic providers currently providing care in humanitarian settings”. 10. By “suspected workload”- do you mean “expected workload”? Please explain. 11. Please elaborate as to what you mean by “expatriate” Do you mean any physician from outside the country of surgical care? Are they from another low resource country or from a high resource country? This in some cases affects the training that they can be expected to have undergone. 12. Change “Missingness” to “Missing data” throughout 13. Please remove unnecessary words such as “specifically”, “additionally”, “However”, “As such”, “therefore”, “Furthermore”, wherever you can. 14. Your conclusion seems to be hinting that because you use various anesthesia providers in a humanitarian setting with MSF equipment and drugs, you can draw conclusions about using various anesthesia providers safely in routine health settings without the checks that MSF provides. It is difficult to draw this conclusion if you don’t know who is actually giving the anesthetic (except in the ND setting). It would also be important to have the peri-operative mortality rates as opposed to intraoperative mortality rates in order to draw conclusions about the safety of the anesthetic care. Patients were largely ASA 1(E) or 2(E) in this study and this may not be the case in routine health settings. You have certainly demonstrated that valuable data can be collected in the most difficult of settings with relative intraoperative safety using a variety of anaesthesia providers in MSF organized settings. This provides a baseline knowledge of what anesthesia care is currently available in humanitarian settings under the conditions described. 15. Thank you for taking the time to use the STROBE and RECORD checklists.
--	---

VERSION 1 – AUTHOR RESPONSE

Reviewer 1 comments:

1. Page 7, Line 30: PA can be clarified better e.g. Physician Anesthesiologist (specialist). It is not clear in this study if there were Physicians/doctors (non-specialists, MD) providing anesthesia? This is

a common scenario in low- and middle-income countries, particularly, 'task sharing' during AC situations, resource limited situations, or unique circumstances as mentioned in page 15, line 42.

2. Page 11, Table 1A: 'MD' has been defined for surgical provider, therefore a similar inclusion (if possible), could be made for anesthesia provider.

Authors' response: Thank you for these comment, and we appreciate the terminology may have been unclear. In the methods section we clarified that within our data (bearing in mind that our data is from only one of five MSF operational centres) the physician anaesthesia providers were exclusively specialist anaesthesiologists (unlike the physician surgical providers).

3. Line 40: NAs are recruited locally (or sent as expatriates if they are senior providers).

Authors' response: The second reviewer also highlighted that the use of "expatriate" can be further elaborated upon. Therefore, we have clarified the above sentence (page 7) to now state "if a project is expected to have a low workload, nurse anaesthetists are either recruited locally or, if they are senior providers, sent over as expatriates from MSF-OCB surgical projects in other countries".

4. Page 12, Line 16: ...and ketamine-based general anaesthesia (GA) without a protected airway. Need to clarify if the above statement is a new variable from what has been identified in Page 30 Appendix, Line 16 Table 1 Variables e.g: General anaesthesia without intubation or muscle relaxant; Other anesthesia.

Authors' response: Thank you, we are referring to the same variable, and have thus clarified it within the manuscript as follows "general anaesthesia (GA) without intubation or muscle relaxant, which for the most part was ketamine-based".

5. Page 13, Table 2: Clarify 'Emergent' (not mentioned anywhere in the appendix).

Authors' response: We have updated the appendix to include the terms used in the tables of the manuscript.

6. Page 16, Line 16, 49: Correct/Rephrase these statements:considering the at times unpredictable nature of working..... 'learning that' could be valuable in many LMIC...

Authors' response: We have altered the wording slightly, to ensure they read better but keep the same meaning. Therefore, they now read: "Data quality is a known issue when using surveillance data, and the occasionally unpredictable nature of working in humanitarian settings means there is a risk of further decline in quality." "This would promote learning on how to optimize the surgical and anaesthetic workforce and help to ensure safe surgical and anaesthetic care, all of which could be valuable in many LMIC routine health settings."

7. Page 19, Line 12, Fig 1: It is not clear as to what is 'Order' under the title Case levels ?

Authors' response: This refers to whether this is the first time the patient has surgery, or whether the patient has been operated upon previously. We have clarified this in the figure, which now reads "Order (first or repeat surgery)".

Reviewer 2 comments:

1. Change the title to make it clearer what you are reporting on. One suggestion might be "Anaesthesia care providers employed in humanitarian settings by Medecins Sans Frontieres: A descriptive study of 173,084 surgical cases over 10 years."

Authors' response: Thank you, we have changed the title to "Anaesthesia care providers employed in humanitarian settings by Médecins Sans Frontières: A retrospective observational study of 173,084 surgical cases over 10 years"

2. Consider rewording your objective. Is your primary objective to characterize the volume and nature of the surgical workload or is that important as context only? What do you mean by exploring the "nature" of the use of different categories of anaesthesia providers? Do you mean "to explore the numbers of different levels of training of anesthesia providers used in humanitarian settings" The objective can be made clearer.

Authors' response: Thank you, the objective has now been altered to reflect the primary focus of the study, and now reads: "The objective of this study is to describe the extent to which different categories of anaesthesia provider are used in humanitarian surgical projects and to explore the volume and nature of their surgical workload." Likewise, in the abstract, the primary objective reads the same, while the primary outcome measure reads "Volume and nature of surgical workload of different anaesthesia providers."

3. In the abstract results there is mention of mortality as "low", please explain compared to what or give the percentage range.

Authors' response: We have now included the percentage data in the abstract.

4. The authors mention natural disasters as one of the settings included in the study but only include the data in the appendix. Consider including it for comparison with the AC and HG settings. Despite the smaller numbers, it is helpful to have it together for comparison.

Authors' response: Thank you for your comment, and it was a consideration as we prepared the tables for the study. However, besides the low number of cases we would suggest keeping it separate in the appendix as the setup is completely different: in the event of natural disasters, MSF has always deployed a full expatriate team, which means adding natural disasters to tables 1 and 2 would provide only limited added comparative benefit, while at the same time making the table more complex. Instead, we have added further detail in the results section, to allow better comparison.

5. Please change the nomenclature. In much of the world "anesthetist or anaesthetist" refers to a non-physician and "anesthesiologist or anaesthesiologist" is a physician. In North America a PA is a physician's assistant. There is a great deal of confusion among many about the difference between a physician and nurse anesthesia provider- even in the medical community. I recommend that you decrease the confusion by using the full name of physician anaesthesiologist where indicated and non-physician anesthetist or non-certified anaesthetist. By "non-qualified" I believe the authors mean non-certified or unofficially trained. I don't think MSF would have permitted someone unqualified (meaning incompetent) to do the work with MSF.

Authors' reply: Thank you very much for this, we have long grappled with the nomenclature and trying to make it reader friendly, so your suggestions are appreciated. We have removed the abbreviated terms (PA, NA, and UA), and have instead written the provider cadre out in full. Furthermore, we agree that "anaesthetist" has different meanings across the globe, and "unqualified" gives the wrong impression. Therefore, throughout the manuscript (and appendix) we now refer to "physician anaesthesiologist", "nurse anaesthetist", and "uncertified anaesthetic provider".

6. The kind of anaesthesia used for cases is very interesting and should have a more prominent place in the article. The authors currently have it in the "Case-level findings".

Authors' reply: Thank you for this comment, we agree this is an interesting topic in its own right. With this article, we aim to provide a descriptive overview of the setting and type of "surgical case" encountered by the different anaesthetic provider types. We believe analysis of anaesthetic technique is more suitably addressed in a separate paper (which we are in the process of writing up), where we

can try and match the cases more evenly between the providers, in order to look at the choice of anaesthesia as a dependent variable.

7. Please say more about the 8 surgical projects that were completely excluded if you know more. It would also be interesting to know about the missing intra-operative mortality data. Which settings did they come from and what else do you know about the providers and case mix from those missing the data? I believe the answers may be in the Matrixplot but this is difficult to read. I wonder if you might try dividing it into two figures to make the labels bigger or excluding data that is irrelevant to the reader such as site ID.

Authors' reply: Apologies, the matrixplot might not add much that couldn't be easier said in words, and for that reason we suggest removing it all together. Instead we have added more detail under the "missing data" paragraph addressing the points you raise (in particular around the missing intra-operative mortality data).

8. Instead of writing "physician-led" you might consider writing "physician available team care". This would make it clearer that you are reporting on care settings where physician anaesthesiologists were available but not necessarily present.

Authors' reply: Thank you, we have tried to emphasise this point through the manuscript, in particular in the discussion and conclusion.

9. I believe your conclusion overstates that you can "inform global debates on the provision of anaesthesia" Perhaps you can "add to the global data on anaesthetic providers currently providing care in humanitarian settings".

Authors' reply: Thank you, the sentence has been changed accordingly.

10. By "suspected workload"- do you mean "expected workload"? Please explain.

Authors' reply: Yes, we mean "expected workload" – changed in the manuscript.

11. Please elaborate as to what you mean by "expatriate" Do you mean any physician from outside the country of surgical care? Are they from another low resource country or from a high resource country? This in some cases affects the training that they can be expected to have undergone.

Authors' reply: For both anaesthesiologists and nurse anaesthetists, the expatriate workforce may be drawn from low income countries, and this is dependent on both the availability and suitability of the individual provider. We have clarified this in the methods section.

12. Change "Missingness" to "Missing data" throughout

Authors' reply: Thank you – we have changed as suggested as on review rather than perform missingness analysis (with for example multiple imputations or other sensitivity analyses) we have essentially described the missing data. As there was only a small proportion of missing data, we do not believe there would be any added benefit from undergoing a deeper level of analysis. Instead we have elaborated on the description of the missing data (as per your point 7), and removed the missingness analysis as a highlight in the "strengths and limitations of the study box".

13. Please remove unnecessary words such as "specifically", "additionally", "However", "As such", "therefore", "Furthermore", wherever you can.

Authors' reply: We have gone through the manuscript and tidied up, thank you.

14. Your conclusion seems to be hinting that because you use various anesthesia providers in a humanitarian setting with MSF equipment and drugs, you can draw conclusions about using

various anesthesia providers safely in routine health settings without the checks that MSF provides. It is difficult to draw this conclusion if you don't know who is actually giving the anesthetic (except in the ND setting). It would also be important to have the peri-operative mortality rates as opposed to intraoperative mortality rates in order to draw conclusions about the safety of the anesthetic care. Patients were largely ASA 1(E) or 2(E) in this study and this may not be the case in routine health settings. You have certainly demonstrated that valuable data can be collected in the most difficult of settings with relative intraoperative safety using a variety of anaesthesia providers in MSF organized settings. This provides a baseline knowledge of what anesthesia care is currently available in humanitarian settings under the conditions described.

Authors' reply: This is a very fair point, and through the discussion we have been keen to emphasise that the study provides a baseline to build upon rather than detailed knowledge on safety of different practitioners, so we appreciate the opportunity to clarify this. Besides the amendment you mention in point 9, we have modified the conclusion to emphasise the need for further study.

15. Thank you for taking the time to use the STROBE and RECORD checklists.

VERSION 2 – REVIEW

REVIEWER	Lena E Dohlman MD MPH MGH/Harvard Medical School USA
REVIEW RETURNED	31-Dec-2019

GENERAL COMMENTS	Anaesthesia care providers employed in humanitarian settings by Medecins Sans Frontieres: A retrospective observational study of 173,084 surgical cases over 10 years General comments: The revised manuscript is much improved. However there are some implied conclusions in the manuscript which do not belong in the paper. Your primary outcome measure was to present the volume and nature of the surgical workload of different anesthesia providers in the setting of MSF-OCB. Your data collection was not designed to determine the mortality outcomes of different providers and yet you emphasize these. There are really too many variables to assign providers to the intra operative mortality rate. As you have pointed out, there is no way to know how involved the physician anaesthesiologists were in the physician led cases. or even if they were present in the hospital at te time the cases were performed. In addition the case mix was different between the different groups- some of which was preselected by assigning less trained staff to less busy facilities. "Physician led" settings were more likely to do trauma cases and nurse led and uncertified (informally trained) personnel were more likely to do obstetrical cases with spinal or ketamine anesthesia. In all settings there were many more cases done in "physician led" areas. Was this because they were pre-selected? The bottom line is that conclusions about the safety of anesthetics given by one group or another cannot be made with the data that you have. Recommendations: Throughout manuscript: Please be consistent in your use of uncertified/unqualified anaesthetic provider in both the written part and tables. You may want to switch it all to "informally trained anaesthetic providers" as you did with "MD" surgical providers. Abstract: In the Results Section- In the last sentence take out the intraoperative mortality figures assigned to individual groups and just include the overall intraoperative mortality figure. Abstract: In Conclusions please change the last sentence to "These data offer a strong foundation to further evaluation of the
---

	use of different models of humanitarian anaesthetic providers in the setting of standardized anaesthetic equipment and medications." Introduction: Line 16- omit "further" and start new sentence with The recognition... Variables and bias: Pg 9, line 8 and 9 is confusing. You could write " While we cannot account for surgical cases not recorded at all, we explored incomplete data that had been excluded, to assess similarity to the included data." Case-level provider findings: Pg 13 lines 5-8. This sentence is misleading. You might change it to " All providers did predominantly non-elective work with trauma surgery more commonly done in physician led projects in both AC and HG and caesarean sections more commonly done in nurse anaesthesia projects, especially in HG settings." You can leave the intraoperative mortality figures in this section since you have adequately explained the selection bias and other variables that make these figures less reliable. Discussion: Pg 16 sentence 42-43. After your comments about all providers taking care of patients with extremes of age and co-morbidities you might point out that a majority of cases were "minor surgeries" which are less risky even with patient with higher ASA class. Do you know how many patients of high ASA class had major surgery and how many minor surgery and in which projects? If not you need to be careful about implying that informally trained anaesthetists can safely take care of high ASA class patients. Conclusion: Pg 17, line 49-51 Consider changing sentence to read "In conflict and healthcare gap settings, nurse anaesthetists and uncertified anaesthesia providers can be used as major providers. This study shows that the humanitarian sector has considerable experience with task sharing and shifting but further study of peri-operative outcomes in these circumstances is needed to draw conclusions about how safe and practical it would be to apply to other settings." Page 18 line 6 Delete "which would be valuable in many LMIC routine health settings." Overall an interesting study which will add to the difficult to collect data on humanitarian mission work.
--	--

VERSION 2 – AUTHOR RESPONSE

Recommendations:

Throughout manuscript: Please be consistent in your use of uncertified/unqualified anaesthetic provider in both the written part and tables. You may want to switch it all to "informally trained anaesthetic providers" as you did with "MD" surgical providers.

Authors reply: Apologies for not changing it in the table that was an oversight. We prefer the term "uncertified" as it emphasises the potentially only difference existing between them and the anaesthetic nurses. We have changed it accordingly.

Abstract: In the Results Section- In the last sentence take out the intraoperative mortality figures assigned to individual groups and just include the overall intraoperative mortality figure.

Authors reply: We agree with your general comments, and would not want to mislead. We have changed it accordingly.

Abstract: In Conclusions please change the last sentence to "These data offer a strong foundation to further evaluation of the use of different models of humanitarian anaesthetic providers in the setting of standardized anaesthetic equipment and medications."

Authors reply: Thank you, the sentence has now been changed.

Introduction: Line 16- omit "further" and start new sentence with The recognition...

Authors reply: Thank you, the sentence has now been changed.

Variables and bias: Pg 9, line 8 and 9 is confusing. You could write " While we cannot account for surgical cases not recorded at all, we explored incomplete data that had been excluded, to assess similarity to the included data."

Authors reply: Thank you for the suggested clarification, changed accordingly.

Case-level provider findings: Pg 13 lines 5-8. This sentence is misleading. You might change it to " All providers did predominantly non-elective work with trauma surgery more commonly done in physician led projects in both AC and HG and caesarean sections more commonly done in nurse anaesthesia projects, especially in HG settings." You can leave the intraoperative mortality figures in this section since you have adequately explained the selection bias and other variables that make these figures less reliable.

Authors reply: Thank you for the suggested clarification, changed accordingly.

Discussion: Pg 16 sentence 42-43. After your comments about all providers taking care of patients with extremes of age and co-morbidities you might point out that a majority of cases were "minor surgeries" which are less risky even with patient with higher ASA class. Do you know how many patients of high ASA class had major surgery and how many minor surgery and in which projects? If not you need to be careful about implying that informally trained anaesthetists can safely take care of high ASA class patients.

Authors reply: Thank you, we have made the clarification as suggested.

Conclusion: Pg 17, line 49-51 Consider changing sentence to read "In conflict and healthcare gap settings, nurse anaesthetists and uncertified anaesthesia providers can be used as major providers. This study shows that the humanitarian sector has considerable experience with task sharing and shifting but further study of peri-operative outcomes in these circumstances is needed to draw conclusions about how safe and practical it would be to apply to other settings." Page 18 line 6 Delete "which would be valuable in many LMIC routine health settings."

Authors reply: Thank you for the suggested clarification, changed accordingly.